# Bioactive Ingredients with Health-Promoting Properties of Strawberry Fruit (*Fragaria* x *ananassa* Duchesne)

**DOI:** 10.3390/molecules28062711

**Published:** 2023-03-17

**Authors:** Joanna Newerli-Guz, Maria Śmiechowska, Alicja Drzewiecka, Robert Tylingo

**Affiliations:** 1Department of Quality Management, Gdynia Maritime University, 81-225 Gdynia, Poland; 2Department of Chemistry, Technology and Biotechnology of Food, Gdańsk University of Technology, 81-223 Gdańsk, Poland

**Keywords:** strawberries, bioactive components, polyphenols, vitamins, organic acids, human health

## Abstract

Strawberries (*Fragaria* x *ananassa* Duchesne) belong to the berry group and are characterized primarily by delightful sensory properties. Due to their chemical composition, these fruits are a rich source of bioactive compounds that can modify the metabolic and physiological functions of the body. The aim of this work is to present the current state of research on bioactive ingredients found in these fruits in the context of their health-promoting properties. The paper presents compiled and reviewed data on the content of polyphenolic compounds, organic acids, and vitamins, especially vitamin C, in strawberries. The content of these compounds is influenced by many different factors that are discussed in the paper. It also draws attention to the presence of oxalates and allergenic compounds, which are classified as anti-nutritional compounds of strawberries.

## 1. Introduction

The strawberry (*Fragaria* x *ananassa* Duchesne) is one of the favorite fruits of consumers in many regions of the world [1]. *Fragaria grandiflora* Ehrh vel *Fragaria* x *ananassa* Duch. is a plant that arose from the crossing of the Chilean variety (*Fragaria chiloensis*) with the Virginian variety (*Fragaria virginiana*), which was done by the French biologist Antoine Nicolas Duchesne in the eighteenth century [2]. Strawberries belong to the *Rosaceae* family, which includes many crops of great economic importance, such as the apple (*Malus domestica*). Strawberries are grown in many countries and different climatic zones, but undoubtedly they are a fruit widely grown in the temperate climate zone. They are among the fruits most often consumed raw [1]. Strawberry fruits are low in calories (32 kcal/100 g) and contain a considerable amount of water (over 80%). At the same time, they provide many bioactive substances, such as vitamins, polyphenolic compounds, fiber, and pectin, as well as mineral substances. Many studies point to their health-promoting properties [3,4,5,6]. Among cultivated strawberries, early, mid-early, mid-late, and late varieties are distinguished. The choice of strawberry varieties in cultivation in different countries is determined not only by weather, climatic, and environmental conditions but also by agricultural culture and tradition, as well as agrotechnical possibilities [7,8,9,10,11,12,13].

## 2. World Production and Consumption of Strawberries

Table 1 provides information on the volume of strawberry production in the world, which has been steadily increasing in recent years [14]. Total strawberry production was about USD 18 billion [15]. Among the producer countries, China and the USA are definitely in the lead, while among European Union countries, Spain and Poland produce the most strawberries [16].

Among the factors responsible for the increase in strawberry consumption is an increase in awareness and education in society. Other factors, such as gender, social status, economic factors, and country or region of residence, also play important roles [18,19]. Consumers prefer fruits with a suitable color (they do not accept green and very dark ones), with a taste similar to ripe strawberry fruits, as well as a pleasant aroma and sweetness [1,20]. Research conducted on shaping the quality of strawberry fruit shows that it takes place to the greatest extent at the pre-harvest stage [12]. Among the factors influencing the development of quality in the pre-harvest period are climatic conditions, agricultural practices, the influence of genetic characteristics, and varieties of strawberries. The role and importance of light intensity and lighting quality on the palatability of fruit and the content of bioactive substances should be emphasized. Research has shown that excess light, especially during droughts, and large shading of crops both significantly affect the quality of fruit [21,22].

Research results also indicate that conventional, integrated or organic farming systems affect the strawberry quality and efficiency, soil quality, as well as the occurrence of powdery mildew and gray mold. Fruit quality, especially dry matter content and texture, is rated much higher for organic fruit, but conventional and integrated plantations are less exposed to powdery mildew and gray mold losses [23,24,25,26,27].

## 3. Health and Nutritional Properties of the Strawberry Fruit

The health properties of the strawberry fruit (*Fragaria* x *ananassa* Duchesne) result from the rich composition of bioactive substances. Thanks to the high water content of over 90% and the low calorific value of 32 kcal/100 g, strawberries are recommended in slimming diets and for fighting obesity [4].

Strawberry fruits, due to the content of polyphenolic compounds and vitamin C, have antioxidant effects. Phenolic compounds, including strawberry phenolic acids, have a wide range of biological activities, from anticancer to anti-inflammatory, neurodegenerative, and antioxidant activities [3,4,28,29] (Figure 1).

Among the various properties of strawberry fruits, attention should be paid to their anti-inflammatory properties and effects on the immune system. Research conducted by Promsong et al. [30] showed that ellagic acid (EA) is responsible for various pharmacological functions of fruits such as pomegranates, blackberries, malines, and strawberries. Studies conducted by these authors on the culture of primary human gingival epithelial cells have shown that EAs found in fruits have an effect on the innate immunity of the oral cavity. Thus, they may play a role in innate mucosal immunity. Favarin et al. [31] also suggested, based on studies conducted on mice, that ellagic acid contained in fruit extracts may be a treatment for reducing inflammation during acute lung injury. In vitro studies have shown that ellagic acid (EA) binds covalently to DNA, and this has been suggested as the mechanism of its antimutagenic action and anticarcinogenic effects [32].

Research by Li et al. (2008) on the effect of strawberry extracts on the epoxide-induced activation of transcription factors and their target genes by the epoxide benzo[a]pyrene diol epoxide (BPDE) suggests that strawberries may target different signaling pathways, exerting antitumor effects [33]. Similar conclusions were made by Somsagar et al. (2012), who administered methanolic strawberry extracts to leukemia (CEM) and breast cancer (T47D) cell lines. They found that the extracts had therapeutic and chemopreventive potential, affected the proliferation of cancer cells by activating apoptosis, and did not cause any side effects. Treatment of mice with breast adenoma using methanolic strawberry extracts blocked tumor proliferation in a time-dependent manner and resulted in a longer life expectancy [34].

## 4. Vitamins in Strawberry Fruits

The vitamins found in the largest amount in strawberry fruits include vitamin C. The role and importance of vitamin C cannot be underestimated. A lot of research and scientific articles have been devoted to this vitamin. Vitamin C plays an important role in many metabolic functions. It is an antioxidant that protects the body from the harmful effects of free radicals and is used as a therapeutic agent in many diseases and disorders. Vitamin C protects the immune system, reduces the severity of allergic reactions, and helps fight infections. The effects of vitamin C on diseases such as cancer, atherosclerosis, type 2 diabetes, neurodegenerative diseases, and metal toxicity are being investigated [35,36,37,38].

The content of vitamin C in strawberries remains in a wide range from 23.16 ± 2.32 [5] to 112.34 mg/100 g [39] (Table 2). This is influenced by many factors, such as climatic and soil factors, fertilization, strawberry variety, plantation age, harvest period, cultivation method (organic, conventional, integrated, hydroponic, tunnel, or in the open field), and storage conditions after harvest [12,40,41]. The vitamin C content in organic strawberries is very often significantly higher than in conventional fruits [42,43]. According to Reganold [44], one serving of strawberries from organic farming provides 9–10% more vitamin C than conventional strawberry fruit.

A comparison of organic and integrated farming systems showed that strawberries from organic cultivation contained significantly more vitamin C [27]. According to previous studies, similar conclusions were made in a meta-analysis conducted by Brandt et al. [53]. However, this higher content is influenced not only by the growing system but also by the variety of strawberries. Kobi et al. [39] conducted research on the impact of the agrosystem of strawberry cultivation of two cultivars, “Camarosa” and “Albion”. The fruits of both these organically grown cultivars contained more vitamin C, but the “Albion” variety contained significantly more vitamin C [27].

The vitamin C content in strawberries is also influenced by the date of harvest. Research by Hallmann et al. [43] showed that strawberries harvested in the third term of the growing season contained significantly more vitamin C. Voća et al. [40] conducted research on the quality of strawberries grown in the hydroponic system, in the tunnel, and in the open field. The highest amount of vitamin C was contained in fruits grown in the high tunnel and ranged from 64.54 mg/100 g to 83.07 mg/100 g fresh weight. The amount of vitamin C was slightly lower in fruits grown in the ground, while the least amount of vitamin C was contained in fruits grown in hydroponics and ranged from 32.42 mg/100 g to 44.97 mg/100 g fresh weight.

Octavia and Choo [51] showed that fruit storage conditions have a major impact on vitamin C content in strawberries. The vitamin C content systematically decreased from the first day of storage in refrigeration conditions at 4 °C, and after 4 days, the decrease in vitamin C was already 55.5%. A 77% decrease in vitamin C content was found by Turmanidze et al. [52] after 8 days of storing strawberries in a refrigerator. In addition, they found that treating the fruit before storage with a solution of 1 or 2% CaCl_2_ had a significant effect on ascorbic acid behavior. This may be because higher concentrations of CaCl_2_ delayed the rapid oxidation of ascorbic acid in the samples. Unfortunately, the authors did not state whether treating the fruit with CaCl_2_ solution affected the sensory experience. The determination of vitamin C content in fruits can be used as an indicator of strawberry freshness because the longer the storage period from harvest, the greater the decrease in the content of this vitamin.

The quality of strawberry fruits intended for transportation or storage is determined by the degree of ripeness. Strawberries for transport are harvested at an incomplete stage of maturity, which is referred to as “pink”. Pineli et al. [48] harvested strawberry fruits in the green, pink, and ripe stages. It was found that the vitamin C content in the pink and ripe stages did not differ significantly.

In the discussion of the influence of various factors on the quality of strawberry fruits, attention was paid to the influence of the environment [46]. According to the authors, the quality characteristics of strawberry fruits, including vitamin C content, can be influenced by interactions between the variety and the environment, which affects the quality and acceptance of the fruit by consumers. Interannual and mid-year variations in the organoleptic and functional fruit quality parameters of five strawberry cultivars over four consecutive growing seasons were analyzed to assess their relative stability. In most cultivars, the organoleptic parameters were characterized by greater interannual stability but greater variability throughout the season, while the performance quality parameters were the opposite. Relative humidity and average and minimum temperatures were partly responsible for variations in fruit quality, but other factors, including genotype, may also have an impact [46]. Strawberry fruit also contains vitamins from other groups, such as A, B, and K. The results of the content of selected vitamins contained in strawberry fruit are presented in Table 3.

Among the B vitamins, attention was paid to the presence of folates. Folate is considered an essential dietary component involved in numerous metabolic pathways, mainly in carbon transfer reactions, such as purine and pyrimidine biosynthesis and amino acid interconversion. Folate has a protective effect against neural tube defects, ischemic events, and cancer. Among other things, a lack of folate results in reduced methionine levels and increased homocysteine levels. The main source of folate is food, especially of plant origin, but it is also synthesized by intestinal bacteria [61,62].

Recent studies have shown that strawberries can be considered a significant source of folate. Table 3 shows that the content of folic acid and total folate can be due to many factors, such as variety, fruit ripeness, and harvest year [59]. It is difficult to compare test results when folate is determined using different methods. Folate content can be determined using a radioprotein-binding assay [58], a microbiological assay [60], or a stable isotope dilution test [59]. However, it can be concluded that regardless of the method of folate determination, fresh strawberries and processed strawberry products are good sources of folic acid. During refrigeration storage, folate is more stable than vitamin C. Research by Rami et al. [57] has shown that storing strawberry fruit in a refrigerator (4 °C) for 14 days resulted in a 28% loss of folic acid.

## 5. The Content of Organic Acids in Strawberry Fruits

Organic acids are the substances that, next to saccharides, have the greatest impact on the palatability of strawberry fruits and their sensory impressions [63]. Many different organic acids have been identified in strawberry fruits, such as malic, tartaric, citric, succinic, oxalic, gallic, and coumaric acids [64,65,66,67]. However, the acids that are found in the largest quantities in strawberry fruits are malic and citric acids. The acid content of strawberry fruits is influenced by many factors. Among the most important are environmental, climatic, and cultivation factors, as well as temperature, light intensity, variety, type and quantity of fertilizers used, water availability, and many more. Cao et al. [64] in fruits of the Hongyan, Tiangxiang, Tongzi, and Zhangji cultivars determined the total organic acid content in a wide range of 874.30–1216.27 mg/100 g fresh weight. They found that the most common acid was citric acid, accounting for 73.5–84.7% of all organic acids. The next most abundant organic acids identified by these authors were malic and oxalic acid, which accounted for 9.5–21.7% and 4.5–7.9% of the total amount of organic acids, respectively. In an earlier study, Skupień et al. [50] reported that malic acid was the most common acid and accounted for 56% of all organic acids in the fruits of the Elsanta variety of strawberries. Ikegaya et al. [65] also determined organic acids, such as citric acid, malic acid, and succinic acid, in strawberry fruits. The level of succinic acid in all cases was below 0.1 g/L and was therefore considered to have a minor effect on taste. In addition, they found that the distribution of organic acids in the fruit is almost uniform throughout the pulp of the strawberry fruit, unlike sugars, which are greatest in the top of the fruit and lowest in the peduncle. This is important when experiencing taste sensations, depending on which part of the strawberry fruit consumption begins. Most often, the top of the strawberry is sweeter, and the peduncle is sourer. 

Factors that significantly affect the composition of strawberry fruits include growing conditions and variety. Gecer et al. [66] evaluated the effect of cultivation in high tunnel and open field conditions, in addition to variety, also influencing on fruit composition. These studies were conducted for the Albion, Kabarla, and Rubygom varieties. The dominant organic acid was malic acid, whose content in the Kabarla variety was 870.729 mg/100 g. 

Factors influencing organic acid content and their importance in strawberries are presented on Figure 2.

Organic acids are also involved in stabilizing the color of strawberry fruits. The color of strawberry fruits is shaped by anthocyanins, whose stability depends on pH [67]. Holcroft and Kader [68] showed the effect of pH and titratable acidity on the color stability of strawberries stored in a controlled atmosphere. Since pH has a profound effect on anthocyanin stability and color expression, especially in aqueous solution, changes in pH can cause significant color loss. Organic acids accumulate in vacuoles and affect the consistency and juiciness of the fruit. It has been noted that the availability of water plays an important role in this process. If it is provided to the plant sparingly, the quantity of organic acids is lower, and the fruits are sweeter. Similar effects were observed in the case of persistent drought [69].

The content of organic acids is important when determining the date of harvesting fruit intended for processing. Research on the influence of organic acids on the palatability and preference of jams obtained from strawberry fruits was conducted by Ikegaya et al. [70]. It has been shown that the intensity of the sweetness of jam decreases with increasing organic acid content. In contrast, the intensity of acidity increases with increasing organic acid content. It has also been observed that the sensation of acidity is influenced by the way the jam is consumed. These impressions changed with the consumption of jam with bread or yogurt. 

Extensive research on the role and importance of organic acids was conducted by Famiani et al. [71]. They analyzed in what period of plant growth and in what part of it the acids accumulate. A characteristic feature of many fruits is that the concentrations of acids increase until the beginning of ripening and then decrease. In turn, in the authors’ own research, the effect of variety of strawberry on organic acid content was determined. Elsanta contained, on average, more malic acid (0.56 ± 0.38 g/100 g d.m.) and citric acid (1.32 ± 0.28 g/100 g d.m.) than Hanoye, in which these contents were 0.36 ± 0.21 and 1.12 ± 0.25 g/100 g, respectively. The results obtained for the average citric and malic acid content were higher than the values presented by Conti et al. [25], regardless of the variety of fruit. In addition, the effect of time of plantation use on organic acid content in strawberry fruits was determined. In general, strawberries harvested in the first year of cultivation contained less malic and citric acids than strawberries harvested in the third year of use of this plantation. Thus, it did not confirm the results of Conti et al. [25], who determined a higher content of citric and malic acid in second-year strawberries than in the first-year plantation.

Ascorbic acid and organic acids also play important roles in the absorption of non-heme iron. Research conducted by Teucher et al. [72] has shown that if a product or food contains the right quantity of organic acids or ascorbic acid, the absorption of iron is more effective. The highest efficiency of absorption of non-heme iron is in the presence of ascorbic acid. The authors believe that it is necessary to further characterize the effectiveness of various organic acids in supporting iron absorption.

Organic acids are also considered by nutritionists in the context of the benefits and risks resulting from their presence in fruits. This is primarily due to the presence of oxalates, which are classified as anti-nutritional compounds. The health risk affects patients with nephrolithiasis and oxaluria [73,74]. The sources of this acid are vegetables and fruits and their products, e.g., spinach, sorrel, rhubarb, coffee, tea, cocoa, sesame seeds, cauliflower, many herbs and spices, beer, and cider [75]. In this case, it is recommended to reduce or eliminate foods that are sources of oxalates in the urine, as well as the simultaneous consumption of foods rich in calcium or supplements that reduce the absorption of oxalate [76].

Strawberries belong to the fruits, which are characterized by a high content of oxalates. Gecer et al. [66] determined three varieties of strawberries in open field crops and in a high tunnel. The oxalic acid content ranged from 39.634 ± 1.61 to 145.373 ± 4.96 mg/100 g for fruits grown in the open field and from 48.789 ± 1.09 to 211.959 ± 6.59 mg/100 g in a high tunnel. 

Other anti-nutritional substances in strawberries are allergenic compounds. They may be responsible for allergic reactions in sensitive individuals, as well as reactions caused by the cross-reactivity of allergens. 

Seven allergenic proteins, including different isoforms, have been found in strawberry fruits. The main allergen in strawberries (*Fragaria* x *ananassa* Duchesne) belongs to the PR-10 group (17 kDa) and is homologous to the major birch pollen Bet v 1 [77]. The Fra a proteins are a major allergen group identified in strawberries [78]. They are members of the pathogenesis-related 10 protein family that causes oral allergic syndrome (OAS) symptoms. Symptoms of OAS include itching, tingling, and swelling in the mouth or throat. Fra a proteins are involved in the flavonoid biosynthesis pathway, which is important for color development in strawberry fruits. Fra a 1 was highly expressed in immature fruit, whereas Fra a 2 was expressed in young to ripe fruit [79]. 

The second group of non-specific lipid transfer proteins includes profilin. It is mainly responsible for strawberry allergies occurring in the Mediterranean area [80,81]. The type of production influenced the allergenic substance content. Aninowski et al. reported that organically produced strawberry fruits are safer because they are less allergenic than conventional and integrated fruits [82].

## 6. The Content of Polyphenolic Compounds in Strawberry Fruits

Flavonoids in strawberries include flavonols, flavanols, and anthocyanins. Flavanols include catechin and epicatechin, while flavonols include kaempferol and quercetin [83]. They play a protective role in carcinogenesis by reducing the bioavailability of carcinogens. Thanks to the presence of flavonoids, strawberries reduce the degree of oxidation of LDL cholesterol.

However, the most important group of flavonoids found in strawberries are the anthocyanins. About 70% of the total antioxidant capacity comes from anthocyanins, highlighting their importance among plant secondary metabolites [4]. Their fruit content increases with the ripening period. The profile composition of anthocyanins in strawberries depends on their genotype. The major anthocyanin in strawberries is pelargonidin 3-glucoside, which has been reported to have anti-inflammatory effects. The other anthocyanins in strawberries are pelargonidin 3-rutinoside and pelargonidin 3-glucoside–succinate [84]. In strawberry skin, the red color is a consequence of anthocyanin biosynthesis and accumulation. In strawberries and all members of the *Fragaria* genus, fruit color is determined by the accumulation of anthocyanins [85].

Anthocyanin accumulates in fruits quickly in the late stages of ripening, beginning when fruits turn from white to red and increasing more than 10-fold in red, ripe berries [86]. They can be practically used as biomarkers in the quality control of products obtained from these fruits because unfair production practices often involve adding other, cheaper fruits to products such as jams and juices. Anthocyanins are a group with exceptionally good scavenging activities [87] and antioxidative and anti-inflammatory properties.

Important phenolic acids in strawberries are the ellagitannins and ellagic acid glucosides, which break down to pure ellagic acid, which is also present in the fruit [88]. Ellagic acid is valuable to human health because it is antimutagenic and has anticarcinogenic activities against chemical-induced cancers [89].

The high level of total antioxidant capacity contained in strawberry fruit enables the neutralization of free radicals and reduces oxidative stress in the human body [29].

The total content of polyphenols in strawberries varies, and many factors affect this parameter. Kolniak [90] defined them at the level of 157.10–178.30 mg/100 g of fresh weight. The content determined by Banaś and Korus [54] was 225 mg TPC/100 g of fresh weight, and by Fijoł-Adach et al. [91], it was 264.00 mg TPC/100 g. The highest TPC content in strawberries was reported by Bojarska et al. at the level of 497.20–787.94 mg TPC/100 g of fresh weight [92].

In the Kashubian strawberries studied by the articles’ authors that were cultivated in three consecutive years, different contents of anthocyanins and total polyphenols were shown, depending on the variety of fruit and the year of harvest. These values ranged from 148.76 to 517.32 mg GAE/100 g of fresh weight for TPC and from 9.79 to 60.28 mg/100 g of fresh weight for anthocyanins [72].

Storage conditions directly affect the nutritional properties of strawberries, including phenolic compounds and free radical scavenging activity [93]. Different storage stages influenced the total content of polyphenols, which have a maximum content of 326 mg GAE/100 g of fresh weight, and a total flavonoid content of 424 mg RE/100 g fresh weight at the end of the refrigeration period [94].

The levels of antioxidants and antioxidant capacity in strawberry extracts from whole fruits vary considerably among genotypes [95]. Cultural systems and different cultivars significantly affected strawberry fruit quality, antioxidant capacity, and flavonoid content. Strawberries grown in the compost sock system had significantly higher flavonoid content and antioxidant capacities than fruit grown in other culture systems [96].

## 7. Conclusions

Strawberries are extremely popular fruits and are valued for their sensory qualities and exceptional palatability. They are characterized by low calorific value and a low glycemic index. They are part of a growing trend that highlights plant-derived antioxidants for their proven health benefits [97].

Strawberries should be consumed primarily fresh. Their fresh consumption makes a real contribution to oxidative status through their high content of phenolic compounds.

However, it should be taken into consideration that strawberries, besides undoubtedly having nutrients, also contain anti-nutrients, which include oxalates. Strawberry consumption should be restricted for consumers with oxaluria, as well as for vegetables and fruits with a high oxalate content, particularly soluble oxalates.

One of the main problems with strawberries is that they are very delicate fruits that are highly susceptible to damage, and they can also suffer post-harvest changes both fresh and during storage.

The sensory qualities of strawberries are primarily due to organic acids and saccharides, which create a specific arrangement and balance in the fruit. Their concentration depends on the harvest period. Strawberry plantation management means capturing the right moment for harvesting fruits that can be intended for direct consumption or processing. Therefore, from the perspective of further research, the authors of this study will deal with the influence of various factors on the quality of strawberries.

## Figures and Tables

**Figure 1 molecules-28-02711-f001:**
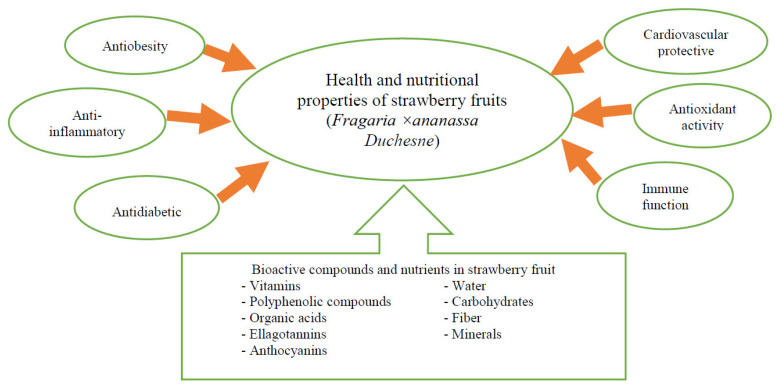
Bioactive compounds and nutrients in strawberry fruit and their impact on human health. Source: self-study.

**Figure 2 molecules-28-02711-f002:**
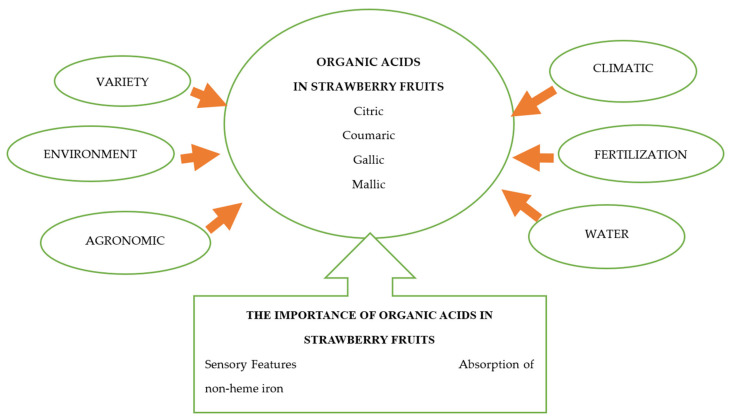
Factors influencing organic acid content and their importance in strawberries (*Fragaria* x *ananassa* Duchesne). Source: self-study.

**Table 1 molecules-28-02711-t001:** World strawberry production.

Country	Production [t]	Production per Person [kg]
China	2,964,263	2.127
USA	1,296,272	3.955
Mexico	653,639	5.24
Turkey	440,968	5.457
Egypt	362,639	3.72
Spain	344,679	7.387
South Korea	213,054	4.126
Russia	199,000	1.355
Poland	195,578	5.089
Japan	163,486	1.292
Morocco	143,440	4.125

Source: [17].

**Table 2 molecules-28-02711-t002:** Vitamin C content in strawberries based on research by various authors.

Vitamin C Content [mg/100 g]	Investigated Factor Influencing Vitamin C Content	Source
63.73 ± 3.98–72.57 ± 3.12	variety	[45]
37.92 ± 0.42–75.50 ± 6.40	variety and environmental conditions	[46]
52.9 ± 0.6–63.4 ± 2.2	agrosystem	[42]
97.93–112.34	agrosystem and variety	[39]
56.6 ± 1.5–62.1 ± 1.5	agrosystem	[44]
28.8 ± 3.7–88.7 ± 8.2	variety	[10]
42.15 ± 2.27–81.62 ± 3.55	variety	[47]
23.16 ± 2.32–52.85 ± 1.03	variety and maturity degree	[48]
81.00–82.50	variety	[49]
54.00–87.00	variety	[50]
32.42 ± 0.71–83.07 ± 0.31	agrosystem and date of harvest	[40]
41.39 ± 9.14–82.64 ± 9.37	date of harvest	[43]
57.00 ± 11	storage condition	[51]
27.35 ± 0.35–45.17 ± 0.24	storage condition	[52]

Source: self-study.

**Table 3 molecules-28-02711-t003:** The content of selected vitamins in 100 g of fresh strawberry fruit.

Vitamin	Content	Source
A [μg]	1.0	[4]
	2.0	[54]
β-Carotene [μg]	25.00 ± 0.02	[54]
	16.00	[55]
E [mg]	0.01	[54]
K [μg]	13.5	[54]
B_1_ [mg]	0.024	[4]
	0.030	[54]
B_2_ [mg]	0.06	[54]
B_3_ [mg]	0.06	[4]
	0.386	[54]
B_4_ [mg]	5.7	[4]
B_5_ [mg]	0.125	[4]
B_6_ [mg]	0.047	[4]
	0.60	[54]
B_7_ [μg]	4.0	[54]
B_9_ [μg]	24	[4]
	75	[54]
5-methyltetraidrofolic acid [μg]	23.57 ± 3.832–237.87 ± 18.932	[56]
	5.286 ± 0.246–6.842 ± 0.317	[56]
Total folate content	90–118	[57]
	335–664	[58]
	59–153	[59]
	20–99	[60]

Source: self-study.

## Data Availability

No new data were created.

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
