# Peer review of "Bioactive Ingredients with Health-Promoting Properties of Strawberry Fruit (Fragaria x ananassa Duchesne)"

_molecules, 2023, doi:10.3390/molecules28062711_

Round 1

Reviewer 1 Report

In this article, the authors have reviewed the Strawberry as a Source of Bioactive Ingredients with Health-Promoting Properties. The authors have compiled and reviewed large number of research studies in these fields and included them in this review. This is an interesting review article in my opinion and can be considered for publication. I have listed some of the modifications that should be incorporated prior to final acceptance.

1.      Reframe the abstract of this article. Write short and good abstract. your abstract looks more introduction than an abstract.

2.      Add some more words in “Key words”.

3.      Write “Health and nutritional properties of strawberry fruit” in detail. Add some more paragraphs (sub-titles) like its role in the immune system, prevention of various diseases, and role as anticancer fruit.  

Author Response

Referee 1

At first thank You for the thorough reviews.

We have modified the manuscript accordingly, and detailed corrections are listed below point by point:

  1. Reframe the abstract of this article. Write short and good abstract. your abstract looks more introduction than an abstract.

ad1.      The correction of abstract was made.

  1. Add some more words in “Key words”.

ad2.      “Key words” were added.

  1. Write “Health and nutritional properties of strawberry fruit” in detail. Add some more paragraphs (sub-titles) like its role in the immune system, prevention of various diseases, and role as anticancer fruit.

ad3.     The correction of the chapter “Health and nutritional properties of strawberry fruit” was made. We did not add subsections because we thought it would be not good for the presented text .

 Hopefully we have addressed all of your concerns.

Best regards Joanna Newerli-Guz               

Reviewer 2 Report

The paper by Newerli-Guz and colleaugues is aimed at reviewing the nutritional role of strawberries in human health.

The paper is well conceived, organized and presented.

I have only minor request:

-       -   The Author state into the abstract that ‘It also draws attention to the presence of anti-nutritional substances such as oxalates and allergenic compounds in strawberries’. However, allergenic compounds, as the proteins belonging to the Fra family, and their implication for human health have not been discussed into the main text. I suggest the Authors to discuss this point.

-         -  Table 2 – the column indicating the ‘Investigated factor influencing vitamin C content’ should be better presented by avoiding the repeating of ‘influence of’ for each line. Only the factors themselves should be indicated, i.e. ‘variety’, ‘agrosystem’, etc..

-          Table3 – I suggest the Authors to list the Vitamins with their precursor/active form in separate sub-rows. For instance, beta-carotene is a precursor of Vitamin A, thus it should be included in the row of Vitamin A; folate content, folic acid and the active form 5-methyltetraidrofolic acid should be listed into the same row. Moreover, Betaine should be excluded from the Table as it is not recognized as a Vitamin.

Author Response

Referee 2

At first thank You for the thorough reviews.

We have modified the manuscript accordingly, and detailed corrections are listed below point by point:

1.The Author state into the abstract that ‘It also draws attention to the presence of anti-nutritional substances such as oxalates and allergenic compounds in strawberries’. However, allergenic compounds, as the proteins belonging to the Fra family, and their implication for human health have not been discussed into the main text. I suggest the Authors to discuss this point.

ad1.We discussed  the presence of anti-nutritional substances such as oxalates and allergenic compounds in strawberries’ in the chapter 5.

  1. Table 2 – the column indicating the ‘Investigated factor influencing vitamin C content’ should be better presented by avoiding the repeating of ‘influence of’ for each line. Only the factors themselves should be indicated, i.e. ‘variety’, ‘agrosystem’, etc..

ad2.Table 2 is corrected due to advise of Referee

  1. Table3 – I suggest the Authors to list the Vitamins with their precursor/active form in separate sub-rows. For instance, beta-carotene is a precursor of Vitamin A, thus it should be included in the row of Vitamin A; folate content, folic acid and the active form 5-methyltetraidrofolic acid should be listed into the same row. Moreover, Betaine should be excluded from the Table as it is not recognized as a Vitamin.

ad3.Table 3 is corrected due to advise of Referee.

Hopefully we have addressed all of your concerns.

Best regards Joanna Newerli-Guz                       

Reviewer 3 Report

I have evaluated the nutrients molecules-2237296 by Newerli-Guz et al. However, before any publication consideration, here are my comments.

1. The title should be:

Bioactive Ingredients with Health-Promoting Properties of Strawberry Fruit: A Concise-Review Study.

2. Abstract: It is necessary to mention the Latin name of the strawberry according to the title used and in line with the purpose of the review being carried out.

3. In the title and aims of this study, there is "Bioactive Ingredients" but unfortunately the authors have not presented data on the metabolites of Strawberry. I suggest providing sub-titles and tables that summarize the metabolites profile of Strawberry obtained from profiling studies with advanced technology such as HPLC/HRMS/NMR etc.

3. Table 3, there needs to be a legend of another name or abbreviation for each vitamin.

4. Table 2, details must be added post-harvest treatment for each studies used, which is suspected to have an effect on vitamin C content.

5. "4. The content of organic acids in strawberry fruits" and "5. The content of polyphenolic compounds in strawberry fruits" would be better presented in the form of a table or an appropriate picture.

Author Response

Referee 3

At first thank You for the thorough reviews.

We have modified the manuscript accordingly, and detailed corrections are listed below point by point:

  1. The title should be:Bioactive Ingredients with Health-Promoting Properties of Strawberry Fruit: A Concise-Review Study.

ad1.Thank You for better title of our work . The correction of the title was made.

  1. Abstract: It is necessary to mention the Latin name of the strawberry according to the title used and in line with the purpose of the review being carried out.

ad2. The correction of abstract was made.

  1. In the title and aims of this study, there is "Bioactive Ingredients" but unfortunately the authors have not presented data on the metabolites of Strawberry. I suggest providing sub-titles and tables that summarize the metabolites profile of Strawberry obtained from profiling studies with advanced technology such as HPLC/HRMS/NMR etc.

ad3.In our opinion we presented bioactive ingredients in strawberry fruits, and we agree with Referee, indeed, there are many articles on metabolites in strawberries using advanced technology. However, due to the enormity of the collected data, we omitted this area of research, the article presented to review is the first in a series, so thank you for suggesting the idea for further work. We changed a little bit the chapter “Health and nutritional properties of strawberry fruit”

  1. Table 3, there needs to be a legend of another name or abbreviation for each vitamin.

ad4. Table3 is corrected

  1. Table 2, details must be added post-harvest treatment for each studies used, which is suspected to have an effect on vitamin C content.

ad5. Table2 is corrected due to advise of Referee.

  1. "4. The content of organic acids in strawberry fruits" and "5. The content of polyphenolic compounds in strawberry fruits" would be better presented in the form of a table or an appropriate picture.

ad6. To the capital "4. The content of organic acids in strawberry fruits" the form of an appropriate picture is added.

Hopefully we have addressed all of your concerns.

Best regards  Joanna Newerli-Guz                                                                          

Round 2

Reviewer 3 Report

Well done